# Micro-dissection and integration of long and short reads to create a robust catalog of kidney compartment-specific isoforms

Hongyang Li[1]⊚, Ridvan Eksi[1]⊚, Daiyao Yi[1], Bradley Godfrey[2], Lisa R. Mathew[3], Christopher L. O'Connor[2], Markus Bitzer[2], Matthias Kretzler[1,2], Rajasree Menon[1,2]*, Yuanfang Guan[1,2]*

1 Department of Computational Medicine and Bioinformatics, University of Michigan, Ann Arbor, Michigan, United States of America, 2 Department of Internal Medicine, University of Michigan, Ann Arbor, Michigan, United States of America, 3 Harvard College, Cambridge, Massachusetts, United States of America

⊚ These authors contributed equally to this work.
* rajmenon@umich.edu (RM); gyuanfan@umich.edu (YG)

**Data Availability Statement:** All data are in the manuscript and/or supporting information files. Data availability 1. Eksi, Ridvan Figshare https://figshare.com/articles/Glom_validated_transcripts_

## Abstract

Studying isoform expression at the microscopic level has always been a challenging task. A classical example is kidney, where glomerular and tubulo-interstitial compartments carry out drastically different physiological functions and thus presumably their isoform expression also differs. We aim at developing an experimental and computational pipeline for identifying isoforms at microscopic structure-level. We microdissected glomerular and tubulo-interstitial compartments from healthy human kidney tissues from two cohorts. The two compartments were separately sequenced with the PacBio RS II platform. These transcripts were then validated using transcripts of the same samples by the traditional Illumina RNA-Seq protocol, distinct Illumina RNA-Seq short reads from European Renal cDNA Bank (ERCB) samples, and annotated GENCODE transcript list, thus identifying novel transcripts. We identified 14,739 and 14,259 annotated transcripts, and 17,268 and 13,118 potentially novel transcripts in the glomerular and tubulo-interstitial compartments, respectively. Of note, relying solely on either short or long reads would have resulted in many erroneous identifications. We identified distinct pathways involved in glomerular and tubulo-interstitial compartments at the isoform level, creating an important experimental and computational resource for the kidney research community.

## Author summary

Gene expression is closely associated with functions of diverse tissues and mechanisms underlying human diseases. The regulation of cellular and physiological functions is further complicated by alternative splicing, which enables a single mRNA to be translated into multiple different protein isoforms. The information from the human genome largely increases during this process. Therefore, it is important to study tissue-specific gene and isoform expression at the microscopic level. In this work, we focused on the kidney, a model tissue that has two compartments with different physiological functions and

fa/7305023 2. Eksi, Ridvan Figshare https://figshare.com/articles/Tubulo_validated_transcripts_fa/7305029 3. https://osf.io/nhkbr/?view_only=6b561d2875f0449fa8e64dc841539780.

**Funding:** This work is supported by NIH R35GM133346 and NSF 1452656 to YG. The funders had no role in study design, data collection and analysis, decision to publish, or preparation of the manuscript.

**Competing interests:** I have read the journal's policy and the authors of this manuscript have the following competing interests: YG receives personal payment from Eli Lilly and Company, which may have sponsored certain aspects of the study to MK.

isoform expressions. Chronic kidney disease is prevalent in 14.5% of the US population. We develop an experimental and computational pipeline for identifying isoforms in glomerular and tubulo-interstitial compartments at the microscopic level. Through analyzing multiple kidney samples by two types of RNA sequencing techniques, we integrated both short reads and long reads to identify transcripts and validate results. We identified a total of 14,739 and 14,259 known transcripts, as well as 17,268 and 13,118 potentially novel transcripts in the glomerular and tubulo-interstitial compartments, respectively. Distinct functional pathways were further revealed in these two compartments.

## Introduction

Gene expression patterns at the microscopic level can facilitate the understanding of the function of a tissue [1]. While single cell sequencing can now provide gene expression levels for individual cells [2], the assignment to respective microscopic structure remains a challenging task. Furthermore, the majority of the experimental protocols use short reads in single cell sequencing. While short reads can provide some information about the isoform expression levels, the assignment is not conclusive.

We aim at addressing the above challenge by developing an integrative protocol to micro-dissect microscopic structures of a tissue, followed by both short-read sequencing and long-read sequencing to identify isoforms, and particularly novel isoforms, in these microscopic structures. We used kidneys as a model tissue, due to both disease relevance and clear delineation of two types of compartments in the kidney: glomerular and tubulo-interstitial compartments. The glomeruli function to filter the blood and extra fluid and wastes pass into the tubule and form urine. Chronic kidney disease is prevalent in 14.5% of the US population [3]. Kidney diseases are traditionally classified based on etiology and pathological appearances, but in reality, kidney diseases are a collection of diseases with unique mechanisms, progression rates, and therapeutic responses despite sharing similar histopathological appearances [4]. Micro-dissected renal biopsy specimens are rich sources for capturing the gene expression data of distinct compartments of the kidney glomerular and tubulo-interstitial compartments. Kidney transcriptomic studies typically use generic transcriptome databases such as Ensembl [5], GENCODE [6], or NCBI [6,7], which are incomplete and miss some of the critical kidney-specific transcripts. Moreover, when we use the complete set of annotated transcripts, some short reads from expressed annotated transcripts may be assigned to very similar non-expressed annotated transcripts. This resulted in miscalculated Fragments Per Kilobase of transcript per Million mapped reads values (FPKM) and power-loss in differential expression analysis, which can be remedied by the long reads. In addition, alternative splicing (AS) exponentially increases the information content of genomes by producing multiple transcripts from a single gene, further complicating the picture. Approximately 95% of multi-exonic human genes undergo AS, producing more than 100,000 distinct transcripts from approximately 20,000 protein-coding genes [8]. Therefore, a complete and reliable database of transcript isoforms that is specifically expressed in the kidney using accurate long reads will improve the characterization of new mechanisms, biomarkers, and therapeutic targets.

Recent progress in single-molecule long-read sequencing has provided powerful new tools for researchers to resolve the previous inaccuracies of short-read RNA or DNA sequencing [9–12]. Pacific Biosciences (PacBio) has developed a technique based on Single Molecule Real Time sequencing (SMRT). The sequencing length of the PacBio RS II platform used in this study is 10 kb, covering the entire size of most eukaryotic transcripts. This capability allows the

sequencing of long sections of genomic DNAs or transcripts without fragmentation or PCR amplification. Thus, PacBio's full-length or nearly full-length transcripts eliminate the need for transcript assembly for downstream analysis. However, one limitation of the PacBio platform is its relatively high sequence error rate, but fortunately, these errors are randomly distributed across the reads. Recently improved read-length and base-calling algorithms of the PacBio SMRT analysis platform and the use of circular molecules have mitigated this error rate. When the read length exceeds the length of the cDNA template, each base pair is covered on both strands multiple times, and these low-quality base calls are aggregated to derive high-quality, single-molecule Reads of Inserts (ROIs). The PacBio long-read transcriptome sequencing platform (Iso-Seq) has been successfully applied to human and other species, and it has shown that the use of Iso-seq has a significant advantage over short-read RNA-Seq methods for identifying novel isoforms, detecting AS and gene fusion events [13–16].

In recent years, many computational methods have been developed to identify isoforms through analyzing RNA-seq reads. For example, Mandalorion was developed to analyze long reads and identify isoforms at the single-cell level in murine B cells [17]. Many expressed genes, including cell surface receptors, were found to display complex isoforms. Later on, FLAIR was developed to analyze full-length transcripts, and study differential splicing events and isoforms in leukemia samples with and without SF3B1 mutation [18]. This work demonstrates the power of full-length isoform analysis in connecting different alternative splicing events in cancer. Recently, TALON was developed as a step of the ENCODE4 pipeline in analyzing long-read transcriptomes [19]. TALON was used in multiple transcriptomes to identify both known and novel transcripts across datasets and platforms. In this study, we examined human kidney cortex tissue to study the overall transcriptome of glomerular (hereafter referred to as "glo") and tubulo-interstitial (hereafter referred to as "tub") compartments using both long reads from PacBio Iso-Seq platform and RNA-seq short reads from Illumina platform. To validate full-length transcripts sequenced by PacBio, we used short reads to generate high-quality sets of transcript isoforms from two distinct clinical sources. Then, we compared the confirmed transcript isoforms to the set of known transcript isoforms in GENCODE and provided the final list of expressed transcript isoforms and their annotation status for researchers in downstream kidney transcriptome studies. With this data collection, we identified a large number of novel transcript isoforms and pathways in these two kidney compartments, creating an important resource for the kidney research community. Most importantly, the experimental and computational pipeline developed in this study is promising to be applied to other tissues to acquire the isoform catalog at microscopic levels.

## Materials and methods

### Ethics statement

The study is approved by the University of Michigan Institutional Review Board (HUM00002468: Expression Analysis in Human Renal Disease). All participants have provided written consent to the study.

### RNA extraction from human kidney cortical tissue

We used healthy human kidney cortex cores obtained from five patients who underwent tumor nephrectomies and 20 healthy samples from the European Renal cDNA Bank (ERCB) study [20]. The kidney cores were immediately placed in RNA. Later solution at 4°C for 12 to 24 hours and then stored at -20°C [20]. Micro-dissection of glomerular and tubulo-interstitial compartments was performed as previously described [21,22].

### RNA library preparation/sequencing using illumina platform

For Illumina RNA-seq runs, TapeStation (Agilent, Santa Clara, CA) assessed the RNA quality while following the manufacturer's recommended protocols. Samples with RNA Integrity Numbers (RINs) of 8 or higher were prepared using the Illumina TruSeq mRNA Sample Prep v2 kit (Catalog #s RS-122-2001, RS-122-2002, Illumina, San Diego, CA) using manufacturer's recommended protocols. 0.1–3μg of total RNA was enriched for mRNA using a polyA purification, and the mRNA was then fragmented and copied into first strand cDNA using reverse transcriptase and random primers. The 3 prime ends of the cDNA were then adenylated, and the adapters were ligated. One such adapter was a six-nucleotide barcode unique to each sample allowing us to sequence more than one sample in each lane of a HiSeq flow cell (Illumina). The products were purified and enriched by polymerase chain reactions to create the final cDNA library. Final libraries were checked for quality and quantity by TapeStation (Agilent) and qPCR using Kapa's library quantification kit for Illumina Sequencing platforms (catalog # KK4835, Kapa Biosystems, Wilmington MA) using manufacturer's recommended protocols. The libraries were clustered on the cBot (Illumina) and sequenced 4 samples per lane on a 50 cycle paired-end for tumor nephrectomy samples, and 1 sample per lane on a 100 cycle paired-end run for ERCB samples on a HiSeq 2000 (Illumina) in High Output mode using version 3 reagents according to manufacturer's recommended protocols.

### RNA sequencing with the PacBio platform

For the PacBio library, equal proportions of RNA from the tumor nephrectomy samples were pooled to form 500 ng of RNA for each compartment and processed for next-generation sequencing (NGS) library preparation. PacBio sequencing library preparation was done according to the manufacturer's recommendation for Isoform Sequencing using the Clontech SMARTer PCR cDNA synthesis kit and BluePippin Size-Selection System. cDNA SMRTbell templates were fractionated into 1 kb– 2 kb, 2 kb– 3 kb, 3 kb– 6 kb, and 5 kb– 10 kb. Sixteen SMRT cells were used in total: two for each size fraction of both glomerular and tubulointerstitial compartments. Sequencing was performed on a Pacific Biosciences PacBio RSII by University of Michigan DNA Sequencing Core. Each glomerular and tubulo-interstitial compartment had eight sequencing cells, generating 132,240 and 125,047 SMRT CFLs respectively. These CFLs were supported by 206,415 and 232,845 SMRT Reads of Inserts (ROIs). These were determined to be full-length based on the apparent 5' and 3' cDNA primer sequences and polyA tail at 3' end.

### Sequence generation and alignments

Illumina RNA-Seq reads were aligned and mapped using STAR [23] version 2.5 to the human genome (hg19 assembly). STAR was run in a 2-pass mode with suggested parameters under "ENCODE options" heading in the STAR manual. Pacific Bioscience SMRT raw reads were initially processed using the Pacific Biosciences SMRT analysis software version 2.3.0. The polymerase reads were partitioned into sub-reads. Read of Inserts (ROI) were generated using the default number of polymerase full passes. The Iso-Seq *classify* tool was then used to separate the ROIs into full-length non-chimeric and non-full length reads. We defined full-length reads as containing 5' and 3' cDNA primers and polyA tails. Then, the Iso-Seq *cluster* tool was used to cluster all the full-length reads derived from the same transcript to produce the consensus full-length transcripts (CFLs). CFLs that were unpolished by Quiver were used in the rest of the analysis because it had been reported that Quiver polishing sometimes obscured the introns [15]. SMRT CFLs were aligned and mapped to the human genome (hg19 assembly) using GMAP [24]. We kept reads mapping to a single location (argument– n 1).

As the next phase of the analysis, the pipeline for identification of transcription start sites, splice junctions and transcription end sites were adapted from the TRIMD pipeline [15,24]. Single-exon CFLs were excluded from the following validation steps, as most of them were potential intronic fragments resulting from pre-processed mRNAs. Single-exon CFLs were added back to analysis before the collapse step.

### Identification of transcription start sites (TSS)

CFL 5' end clusters were generated with CFL 5′ ends mapping within 8 bp of each other. Only CFL isoforms whose 5' ends did not contain mismatches were used. A single TSS is determined for each of the clusters by calculating weighted (based on the number of SMRT reads for each start coordinate) averages of the start coordinates of CFLs within the cluster. These consensus TSS are considered validated if there is an annotated transcription start site within 10bp vicinity [24]. Annotated TSS are extracted from GENCODE comprehensive annotation set (version 24).

### Identification of splice junctions

Splice junctions from Iso-Seq CFLs were identified using GMAP, and splice junctions for Illumina reads were identified with STAR [23]. Splice junctions from Iso-Seq CFLs are required to have at least 1 full-length read spanning it to be identified. A splice junction from an Iso-Seq CFL is marked as validated if at least 3 short reads are spanning it or if the junction is already annotated. Annotated junctions are extracted from GENCODE comprehensive annotation set (version 24).

### Identification of transcription end sites (TES)

Illumina reads that have poly(A) tails were extracted from SAM alignment files. These putative reads with poly(A) tails are the reads that have a FLAG code as being first-of-pair, and either end with a run of at least five As, at least two of which are mismatched on plus strand or that start with a run of at least five Ts, at least two of which are mismatched on minus strand. The alignment position base next to the mismatched location was considered a candidate TES. TES that are within 8 bp of each other were considered single candidate TES. The consensus TES coordinate was determined using a weighted average of putative 3′ ends based on a few short reads supporting each TES coordinate. TES are marked as validated either if there is an Illumina TES on the same strand within four bases upstream or ten bases downstream or if there is an annotated TES in the 10bp vicinity [15].

Iso-Seq CFL 3′ ends that align within 8 bp of each other on the genome are considered a single candidate TES. The CFL consensus TES were determined by calculating weighted averages of the end coordinates. Weights are determined by the number of PacBio consensus sequence reads ending at each coordinate. Only putative PacBio 3' end sites that are supported by at least three SMRT reads are kept.

### CFL validation and comparison to known annotations

TSS, splice junction and transcription end sites of each glomerular CFLs are compared to coordinates extracted from short-reads from ERCB glomerular samples and annotated transcripts. For tubulointerstitial compartment, transcript features are compared to short-read RNA-seq data from matching tumor nephrectomy samples, and ERCB tubulo-interstitial samples separately. Iso-Seq CFL validation was done by validating every splice junction present in the CFL. An Iso-Seq CFL was considered validated if every splice junction was validated based

on the criteria explained above. Two different short-read RNA-seq datasets for the tubulo-interstitial compartment were combined for this step. Single-exon CFLs were added to the set of validated multi-exon CFLs, pending further investigation based on their relative location to a known transcript. The set of validated multi-exon CFLs and single-exon CFLs were collapsed with the collapse_isoforms_by_sam.py script in the tofu package provided by PacBio, which is the developmental version of the official Iso-Seq protocol. Then, collapsed isoforms were compared to the GENCODE comprehensive set of annotations (version 24) with a cuffcompare tool from Tuxedo suite of tools [25–28].

## Data records

The set of validated and collapsed isoform sequences with their corresponding annotation class based on cuffcompare tool can be found in "Glom_validated_transcripts.fa" (Data Citation 1) for glomerular compartment and in "Tubulo_validated_transcripts.fa" (Data Citation 2) for tubulo-interstitial compartment. Both files are in fasta format. The ID line for each sequence entry contains an internal transcript ID (PB.X.X), chromosome name, location, cuffcompare class code and associated gene's ENSEMBL gene id.

## Technical validation

For short-read RNA-seq runs, only samples with 8 or higher RINs were used. For long-read RNA-seq with the PacBio RS II platform, equal proportions of RNA from the tumor nephrectomy samples were pooled to form 500 ng of RNA for each compartment and processed for next-generation sequencing (NGS) library preparation. Throughout this study, every sample preparation step was done according to the manufacturer's recommendation.

# Results

## Long-read sequencing analysis of kidney transcriptome

In this study, we micro-dissected glomerular and tubulo-interstitial compartments from 25 healthy human kidney cortex core samples following [21,22] (detailed protocol please see **Materials and Methods**). Among the 25 samples, five were from patients who underwent tumor nephrectomies and 20 healthy samples from the European Renal cDNA Bank (ERCB) study [20]. For the Illumina RNA-seq short reads, we followed TapStation's standard protocols to assess RNA quality. We checked the quality of the final cDNA libraries using Kapa's library quantification kit following the manufacturer's recommended protocols. Then the short reads were mapped to the human genome (hg19) using STAR version 2.5 in a 2-pass mode following the standard "ENCODE options" in the STAR manual. For the PacBio long reads, PacBio's Iso-seq protocol was used for library preparation and long-read sequencing, and PacBio's SMRT Portal was used for the initial data analyses. We processed them using SMRT version 2.3.0 and generated reads of inserts using the default parameters of polymerase full passes. The consensus full-length transcripts (CFLs) were also mapped to hg19 using GMAP. About 98% of the CFLs were uniquely mapped for both kidney apartments. Each glomerular and tubulointerstitial compartment had eight sequencing cells, from which we generated 132,240 (glo) and 125,047 (tub) SMRT CFLs, validated by two high-quality sets of transcript isoforms from two distinct clinical sources based on short reads (**Fig 1**). These were determined to be full-length based on the apparent 5' and 3' cDNA primer sequences and polyA tail at 3' end. Glomerular CFLs ranged from 300 to 27,890 bases in length with a mean of 2,446 bases. Tubulointerstitial CFLs ranged from 300 to 22,765 bases in length with a mean of 2,862 bases. Size fractionation of the library before sequencing reduced the bias toward shorter transcripts.

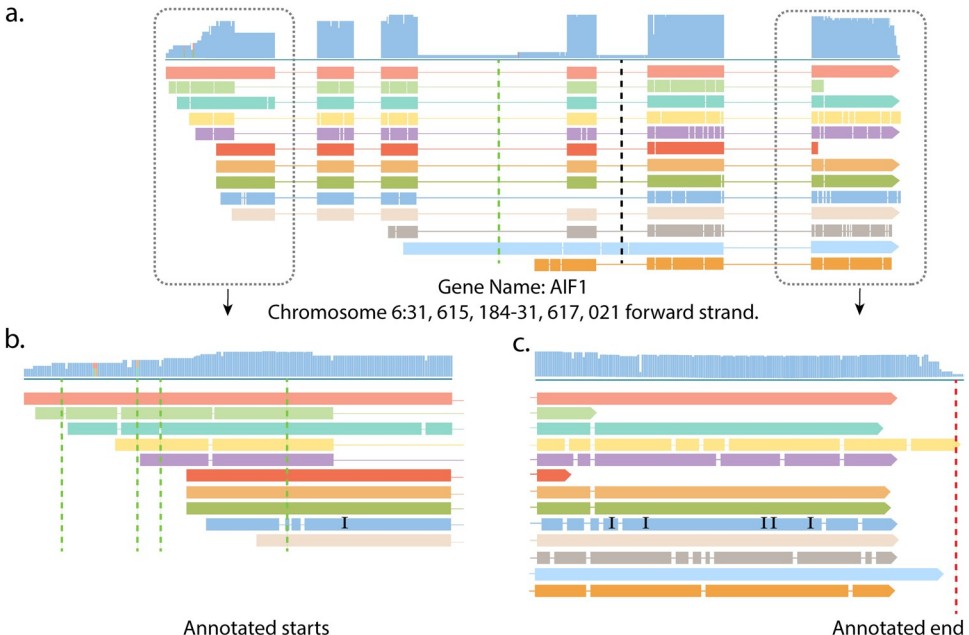

**Fig 1. Overall study design.** The features of every multi-exon Consensus Full-Length transcripts (CFL's) found in PacBio reads were validated through Illumina short reads from two different RNA-seq datasets. Whole CFLs were then validated at every splice junction before comparison to GENCODE annotation.

We further identified 42,896 and 38,831 putative transcription start sites (TSS) from PacBio glomerular and tubulointerstitial multi-exon CFLs respectively (see **Materials and Methods**). As demonstrated in **Fig 2A and 2B** using sample gene AIF5, we could not reliably infer TSS from Illumina short reads; our validation criteria relied only on annotated TSS: Top tracks in the **Fig 2A–2C** are illumina short reads. Just by looking at the read coverage of illumina reads, we cannot pinpoint a location for the TSS, meaning there is not a cliff where read coverage suddenly starts. As a result, we only used TSS that exists in the annotation file. Gene AIF5 has multiple CFLs with varying start positions. During transcript collapsing the CFLs that are identical other than their start positions are merged, and the start coordinate of CFL with the longest 5' end is taken as the true 5' end for the merged transcript. For gene AIF5, we identified 13 TSS and 6 of them were more than 10bp away from the annotated TSS.

**Fig 2. Variation of transcription start and end positions of consensus full-length transcripts.** (A) Thirteen multi-exon consensus full-length transcripts (CFL's) from gene AIF1 locus. (B) The first exon with vertical (green) lines demonstrating the location of annotated transcription start sites (TSS). Thirteen CFL's have a total of 13 different TSS, only 6 out of 13 TSS are within 10 bp of an annotated TSS. (C) The last exon with vertical (red) line shows the only annotated transcription end sites (TES). Thirteen CFL's have a total of 7 different TES, only 4 out of 7 TES are within 10 bp annotated TES.

**Table 1. Validation of PacBio transcript features from kidney tissue.**

| | TSS | Splice Junctions | TES | Validated Multi-exon CFLs |
|---|---|---|---|---|
| **Glomerular** | | | | |
| PacBio with Illumina ERCB | | | | |
| Annotation only | 3732 | 1194 | 4850 | 47785 |
| Short reads only | | 5583 | 39 | |
| Short reads or annotation | | 111506 | 5071 | |
| Total | 38831 | 171742 | 26260 | 71492 |
| **Tubulointerstitial** | | | | |
| PacBio with Illumina TN | | | | |
| Annotation only | 3534 | 7329 | 5278 | 53540 |
| Short reads only | | 3105 | 8 | |
| Short reads or annotation | | 119932 | 5318 | |
| PacBio with Illumina ERCB | | | | |
| Annotation only | | 516 | 4883 | |
| Short reads only | | 8824 | 104 | |
| Short reads or annotation | | 125651 | 5425 | |
| Total | 42896 | 185185 | 24625 | 75573 |

TSS- transcription start site; TES- transcription end site; CFL consensus full-length transcripts; ERCB European renal cDNA bank; TN tumor nephrectomy.

Through mapping PacBio multi-exon CFLs to the genome, we identified 171,742 and 185,185 splice junctions in glomerular CFLs and tubulo-interstitial CFLs, respectively. Each splice junction has at least one full-length SMRT read spanning them. We further utilized short reads to validate the new splice junction findings. A validated set of splice junctions is the union of annotated junctions and splice junctions that have at least three short reads spanning them from Illumina RNA-Seq data. For glomerular compartments, approximately 65% of PacBio junctions were validated. In tubulo-interstitial compartments, using the shallower tumor nephrectomy Illumina RNA-seq data, we validated approximately 64% of all PacBio tubular splice junctions. For example, for tubulo-interstitial, the deeper ERCB RNA-seq data allowed us to validate an additional 8,824–3,105 = 5,719 novel splice junctions (**Table 1**).

We further identified 24,625 and 26,260 putative transcription end sites (TES) in PacBio glomerular and tubulo-interstitial multi-exon CFLs through the process explained in **Materials and Methods**. Then, we extracted short reads containing polyA reads from Illumina RNA-Seq data and extracted polyA site coordinates from those reads. We considered putative TES from PacBio validated if they were near either an annotated TES or a polyA site extracted from short reads. Although it is not as prevalent as the 5' ends, we have a high variation in the transcript end sites extracted from PacBio CFLs (**Fig 2C**), and this variation is incompletely captured by short-reads as we have a limited number of reads with polyA tails. We decided to provide CFLs present in the sample with their original 3' end locations to give readers a choice to process CFLs with different 3' UTR lengths based on their study objective.

## CFL validation and collapsing into final structures

Since exact TSS and TES coordinates do not agree well with annotated coordinates, we did not use these transcript features in whole CFL validation. Our validation criterion required the CFL to have all its junctions validated either by short read support or by annotation. With this criterion, 53,540 tubulointerstitial multi-exon CFLs and 47,785 glomerular multi-exon CFLs were marked as validated. Any two isoforms that differed on the 3' end by more than 100 bp (a

defined threshold by PacBio) were considered different isoforms. If two isoforms differed only by their 5' ends, meaning one isoform had 0, 1, or more 5' exons than the others but all remaining exons agreed, then the shorter isoform was considered identical to the longer one, and it was collapsed into the longer isoforms. After collapsing, 45,778 and 58,378 isoforms were formed from tubulointerstitial and glomerular tissue.

## Comparison of validated isoforms to annotated transcripts

We compared the list of collapsed CFLs to the annotated set of transcripts from GENCODE version 24. The comparison was made with the cuffcompare tool in Cufflinks which classifies each input transcript into twelve distinct classes based on their overlap with an annotated transcript. The seven most prevalent classes and the numbers of collapsed transcripts from tubulo-interstitial and glomerular compartments belonging to each of these seven classes are shown in **Table 2**. "Complete match of intron chain with an annotated transcript," and "Contained within a reference transcript" classes comprise the set of expressed annotated transcripts in the sample. The set of transcripts that belonged to "A transflag falling entirely within a reference intron" class was discarded. These are single exon transcripts, which are most likely by-products of the intron decay process [13]. Other single exon transcript classes are "Single exon transfrag overlapping a reference exon and at least 10 bp of a reference intron" and "Exonic overlap with reference on the opposite strand class", which were also discarded. These are possible pre-mRNA fragments that were pulled down due to the inefficient polyA selection step. The class of "Potentially novel isoforms" includes transcripts that share at least one junction with an annotated transcript and have junctions that do not occur in any annotated transcript. If junctions occur in the annotated transcript, their combination is novel. Again, readers should be cautious about the exact location of TSS in these transcripts as transcripts may be truncated.

The class of "Intergenic transcripts" includes transcripts that map to an intergenic location. Most of these intergenic transcripts are single exon transcripts, more likely regulatory non-coding mRNAs. The two sets of validated and collapsed isoform sequences with their corresponding annotation class are provided.

## Example novel isoforms and novel intergenic transcripts identified by Pacbio long-read sequencing

NPHS2 (podocin) is a protein-coding gene located on chromosome 1. This gene has 8 exons and has two protein coding splice variants according to the GENCODE database. NCBI's

**Table 2.  Classification of validated-collapsed isoforms to GENCODE annotation using the Cuffcompare tool.**

| | Type of Match | Validated Tubulo—interstitial Isoforms | Validated Glomerular Isoforms |
|---|---|---|---|
| 1 | Complete match of intron chain with an annotated isoform | 10407 | 10882 |
| 2 | Contained within a reference isoform | 3852 | 3857 |
| | Total annotated transcripts | 14259 | 14739 |
| 3 | Potentially novel isoform | 8910 | 7767 |
| 4 | Intergenic transcript | 4208 | 9501 |
| | Total novel transcripts | 13118 | 17268 |
| 5 | A transfrag falling entirely within a reference intron | 11407 | 16627 |
| 6 | Single exon transfrag overlapping a reference exon and at least 10 bp of a reference intron | 3729 | 4956 |
| 7 | Exonic overlap with reference on the opposite strand | 2310 | 3420 |

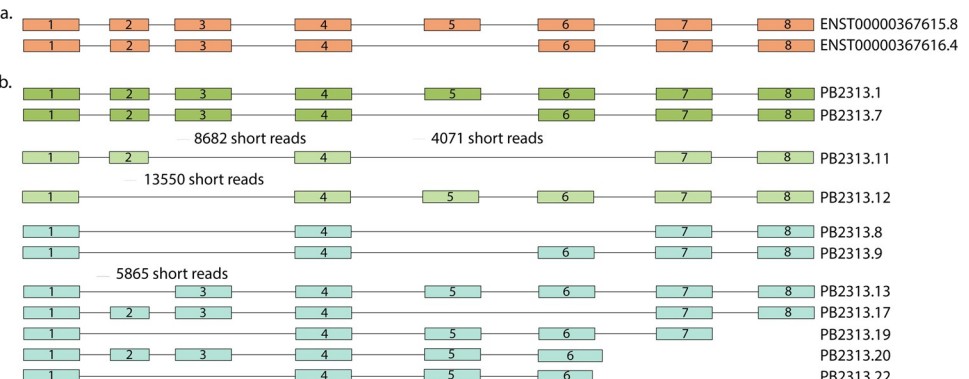

**Fig 3. Gene model for podocin gene NPHS2. (a)** Transcript structures of two annotated transcripts of NPHS2 in GENCODE. The second shorter annotated transcript is missing exon number 5. **(b)** Transcripts found in the PacBio glomerular sample and validated by our method. First two isoforms (dark green) match exactly to the annotated transcripts in GENCODE. Next two isoforms (light green) match two predicted transcript variants in NCBI's RefSeq annotation (XM_017002298.1 and XM_005245483.3). The remaining seven isoforms (gray) isoforms are potential novel transcript variants of NPHS2. The number of uniquely mapped reads to each of the novel junctions are noted above junctions.

RefSeq lists three other predicted protein-coding splice variants. Our validated set of glomerular isoforms has 11 different splice variants for this gene (**Fig 3**). Two of these precisely match variants in GENCODE, and the other two match to the two predicted splice variants in RefSeq. Four other splice variants have the same first and last exons. However, their different combinations make them novel variants. The remaining three novel splice variants have alternative end sites. Among the seven novel splice variants, there is one novel junction, which is supported by multiple short reads.

Among the final set of isoforms, there are 4,208 and 9,501 intergenic transcripts from tubulo-interstitial and glomerular compartments. The majority of these transcripts have single exons, and therefore their junctions do not require validation. There are 76 and 55 multi-exon intergenic transcripts in tubulointerstitial and glomerular compartments. Because they passed the validation criteria, all the junctions in these transcripts are supported by multiple short reads.

## Comparison of the expressed set of annotated transcripts in glomerular and tubulo-interstitial compartments

Glomerular and tubulo-interstitial compartments are expected to have distinct transcriptome profiles. In this section, we compared the set of expressed annotated transcripts from each compartment. 3,993 and 13,536 annotated transcripts are expressed in these compartments, of which 8,198 are common transcripts (**Fig 4A**). We performed KEGG pathway enrichment on the transcripts that are uniquely expressed in each compartment. A full list of enriched pathways is in S1 Table. **Fig 4B** shows the top 3 enriched pathways for each compartment. Glomerular-only expressed transcripts are enriched for Non-alcoholic fatty liver disease (NAFLD), RAP1 signaling pathway, and ubiquitin-proteosome pathway. Tubulointerstitial transcripts are enriched in metabolic, ribosomal, and aldosterone-induced sodium reabsorption pathways.

## Discussion

Recent advances in computational methods have shown great promise to unveil biological insights underlying the complex alternative splicing events and isoforms in a variety of cell

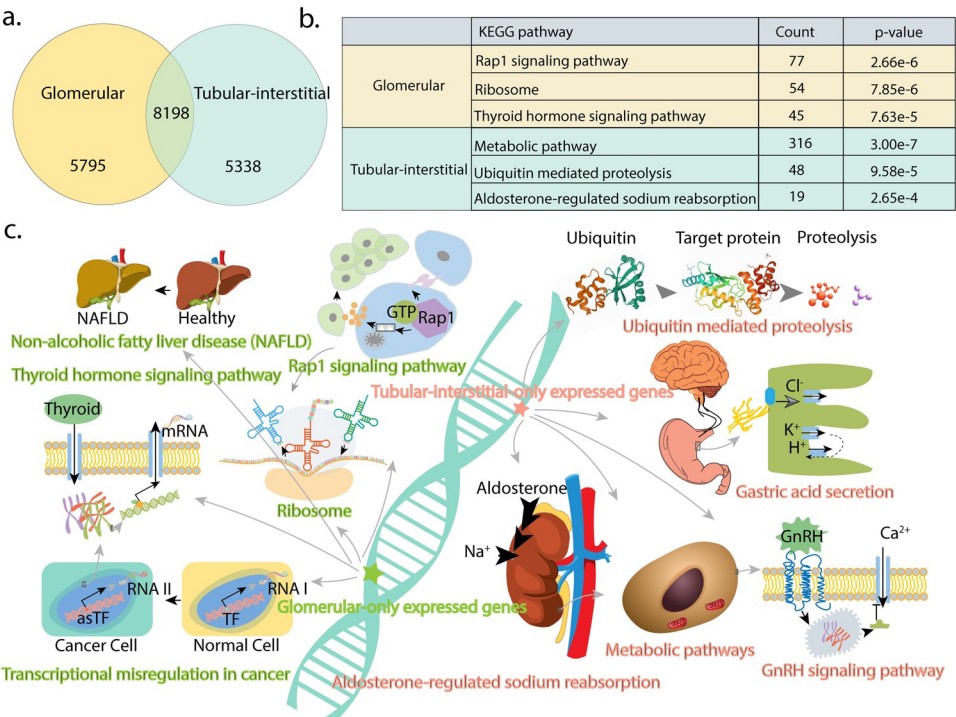

| | KEGG pathway | Count | p-value |
|---|---|---|---|
| Glomerular | Rap1 signaling pathway | 77 | 2.66e-6 |
| | Ribosome | 54 | 7.85e-6 |
| | Thyroid hormone signaling pathway | 45 | 7.63e-5 |
| Tubular-interstitial | Metabolic pathway | 316 | 3.00e-7 |
| | Ubiquitin mediated proteolysis | 48 | 9.58e-5 |
| | Aldosterone-regulated sodium reabsorption | 19 | 2.65e-4 |

**Fig 4. Annotated transcripts and enriched KEGG pathways. (a)** Venn diagram of expressed annotated transcripts from glomerular and tubulointerstitial compartments. **(b)** Top three KEGG pathways and corresponding p-values for the set of transcripts enriched only in glomerular and tubulointerstitial compartments. **(c)** Illustrated pathways enriched in the two compartments of the kidney.

types and tissues, such as Mandalorion [17], FLAIR [18] and TALON [19]. To address the specific situation in our work, we developed a unique in-house pipeline to integrate both long and short RNA-seq reads across individuals. This pipeline enables us to validate reported isoforms, as well as discover new isoforms in two kidney compartments. In this study, we aim at exploring the potential of using combined information from long- and short-read sequencing information to create catalogs of expressed isoforms for microscopic structures. Towards this goal, we used human kidney tissues, which consists of structures of distinct physiological functions. By micro-dissecting glomerular and tubular compartments, we generated biosamples whose half goes to short-read sequencing and the other half goes to long-read sequencing. We designed a pipeline to first use long reads to identify all putative isoforms, and use the short-reads to confirm their relevant junctions. This approach is distinct from current single-cell sequencing efforts or traditional bulk sequencing, in that a microscopic structure carrying out specific physiological function is the study target.

We found tens of thousands of novel transcripts in glomerular and tubular compartments validated by both long and short read sequencing, creating a rich repertoire of transcripts for future functional studies. Preliminary analysis of these transcripts demonstrates drastically different pathway enrichment between the glomeruli and the tubular compartment, supporting the success of this approach.

We envision several future directions of investigation. First, such experimental protocols of microscopic investigation of isoforms can be carried out in other tissues. Second, many novel transcripts are identified in the study, connecting them to isoform function prediction methods, *e.g.*, [29] will help us to elucidate the activated pathways of these isoforms. Third,

connecting the expression patterns from these microscopic structures to single-cell level sequencing can help us understand the variability of the expression patterns of these isoforms. Fourth, the focus of this paper is on health tissues. It will be interesting to study the isoform changes between the healthy state and the disease state of the same microscopic structures. We foresee all these will be exciting opportunities when such microscopic dissection followed by isoform identification with long and short-reads will become well accepted in the medical field.

## Supporting information

**S1 Fig. Distribution of the lengths of the consensus full-length transcripts.**
(TIF)

**S1 Table. Compartment-specific genes expression enrichment analysis.**
(XLSX)

## Author Contributions

**Conceptualization:** Ridvan Eksi, Matthias Kretzler, Yuanfang Guan.

**Data curation:** Ridvan Eksi, Bradley Godfrey, Christopher L. O'Connor, Markus Bitzer, Rajasree Menon.

**Formal analysis:** Hongyang Li, Ridvan Eksi.

**Funding acquisition:** Yuanfang Guan.

**Supervision:** Yuanfang Guan.

**Visualization:** Hongyang Li, Ridvan Eksi, Daiyao Yi.

**Writing – original draft:** Hongyang Li, Ridvan Eksi, Lisa R. Mathew.

**Writing – review & editing:** Hongyang Li, Yuanfang Guan.

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
