## [Decision Letter · Decision Letter 0]

24 Oct 2021

Dear Dr. Guan,

Thank you very much for submitting your manuscript "Micro-dissection and integration of long and short reads to create a robust catalog of kidney compartment-specific isoforms" for consideration at PLOS Computational Biology.

As with all papers reviewed by the journal, your manuscript was reviewed by members of the editorial board and by several independent reviewers. In light of the reviews (below this email), we would like to invite the resubmission of a significantly-revised version that takes into account the reviewers' comments.

We cannot make any decision about publication until we have seen the revised manuscript and your response to the reviewers' comments. Your revised manuscript is also likely to be sent to reviewers for further evaluation.

Sincerely,

Katalin Susztak

Guest Editor

PLOS Computational Biology

Ilya Ioshikhes

Deputy Editor

PLOS Computational Biology

Reviewer's Responses to Questions

**Comments to the Authors:**

Reviewer #1: In this paper, the authors used both PacBio Iso-Seq long reads and Illumina RNA-seq short reads’ information to analyze and identify expressed isoforms for defining microscopic structures. They introduced a pipeline, in which they used long-reads of kidney tissues from two compartments: glomerular and tubule-interstitial and validated the reads by RNA-seq reads of the same tissue. From their experiment, they identified a substantial number of transcription start sites (TSSs), transcription end sites (TESs) and Splice junctions from a filtered set of high quality transcripts.

The manuscript is well written and easy to understand. The figures and tables are also clear and elaborate. However, I have some concerns about the work:

Major:

1. The novelty of the study is limited. The authors use the existing methods to analyze the Iso-seq and RNA-seq data, and identify expressed annotated and unannotated isoforms. The contribution to the computational biology field is not clear. This study can be easy applied to other tissue samples with both matched RNA-seq and Iso-seq data.

2. hg19 annotation is quite old (2009). It would be better to run the experiments based on the latest hg38 annotation.

Minor:

1. Abstract line 2: insterstitial -> interstitial

2. Abstract line 5: microdissed -> microdissected?

Reviewer #2: The paper titled “Micro-dissection and integration of long and short reads to create a robust catalog of

kidney compartment-specific isoforms” describes an developed experimental and computational pipeline for identifying isoforms at microscopic structure-level. It applies Pacific Biosciences SMRT analysis software and Illumina reads approach to discover novel transcripts. Although it is a promising approach, current manuscript lacks details and interpretations.

1. The authors claimed that they developed the approach for isoform identification. However, as far as I know, there are some additional methods that have been developed recent years, such as Mandalorion (Byrne et al. Nat. Comm. 2017) FLAIR (Tang et al, Nat. Comm. 2020), TALON (Wyman et al. biorxiv). I do not mean that they need to compare to all existing methods, but it is important that a comparison is performed to demonstrate the performance of this developed approach.

2. In general, it would be better to describe with more details of read correction, transcript assembly, and transcript quantification in the results part of the manuscript to illustrate the power of this approach and the reason that this approach performs good, such as what are the advantages and what are the drawback of this approach.

3. To test the accuracy of this approach, it would be better to provide the rate of false discovered isoforms and illustrate the reason of these false discovered isoforms.

4. For enrichment results, it would be better to show P-values, gene/transcript count for each pathway, and top pathways in one Figure.

5. The Introduction and Discussion sections are not comprehensive and do not present readers a view of the field. The authors should expand it and describe what are already available and what are the specific features of existing methods for PacBio data. Some recently published methods on novel isoform discovery are not cited. While I understand that this paper focuses on application of PacBio data, it is still important to summarize state-of-the-art methods to present a comprehensive view of the current state of art.

6. The font size in Figures is too small to read

**Have the authors made all data and (if applicable) computational code underlying the findings in their manuscript fully available?**

Reviewer #1: **No: **Illumina RNA-seq and PacBio Iso-seq of the 25 samples are not provided.

Reviewer #2: Yes

PLOS authors have the option to publish the peer review history of their article (what does this mean?). If published, this will include your full peer review and any attached files.

Reviewer #1: No

Reviewer #2: No
---

## [Decision Letter · Decision Letter 1]

19 Mar 2022

Dear Dr Yuanfang Guan,

We are pleased to inform you that your manuscript 'Micro-dissection and integration of long and short reads to create a robust catalog of kidney compartment-specific isoforms' has been provisionally accepted for publication in PLOS Computational Biology.

Best regards,

Katalin Susztak

Guest Editor

PLOS Computational Biology

Ilya Ioshikhes

Deputy Editor

PLOS Computational Biology

No further comments

Reviewer's Responses to Questions

**Comments to the Authors:**

Reviewer #1: The authors have responded well to the previous critiques.

**Have the authors made all data and (if applicable) computational code underlying the findings in their manuscript fully available?**

Reviewer #1: None

PLOS authors have the option to publish the peer review history of their article (what does this mean?). If published, this will include your full peer review and any attached files.

Reviewer #1: No

---

## [Editor Report · Acceptance letter]

11 Apr 2022

PCOMPBIOL-D-21-01107R1 

Micro-dissection and integration of long and short reads to create a robust catalog of kidney compartment-specific isoforms

Dear Dr Guan,

I am pleased to inform you that your manuscript has been formally accepted for publication in PLOS Computational Biology. Your manuscript is now with our production department and you will be notified of the publication date in due course.

With kind regards,

Olena Szabo
